# GENDEXHAND: GENERATIVE SIMULATION FOR DEXTEROUS HANDS

## ABSTRACT

Data scarcity remains a fundamental bottleneck for embodied intelligence. Existing approaches use large language models (LLMs) to automate gripper-based simulation generation, but they transfer poorly to dexterous manipulation, which demands more specialized environment design. Meanwhile, dexterous manipulation tasks are inherently more difficult due to their higher degrees of freedom. Massively generating feasible and trainable dexterous hand tasks remains an open challenge. To this end, we present **GenDexHand**, a *generative simulation pipeline* that autonomously produces diverse robotic tasks and environments for dexterous manipulation. **GenDexHand** introduces a closed-loop refinement process that adjusts object placements and scales based on vision-language model (VLM) feedback, substantially improving the average quality of generated environments. Each task is further decomposed into sub-tasks to enable sequential reinforcement learning, reducing training time and increasing success rates. Our work provides a viable path toward scalable training of diverse dexterous hand behaviors in embodied intelligence by offering a simulation-based solution to synthetic data generation. Our anonymous website: https://sites.google.com/view/gendexhand.

## 1 INTRODUCTION

A long-term goal of artificial general intelligence lies in the development of embodied agents capable of interacting with the real world under autonomous control. Robot learning at scale is particularly promising, as it holds the potential to endow agents with the breadth of skills and robustness necessary for complex real-world deployment (Intelligence et al., 2025; Black et al., 2024; Liu et al., 2025). Large-scale, high-quality data emerges as a cornerstone for effective robot learning, particularly in manipulation, where diverse datasets drive improvements in policy robustness and generalization (Torne et al., 2024; Lin et al., 2025; Ai et al., 2025). However, constructing complex and diverse simulation environments or collecting data with real-world robotic platforms is both costly and technically challenging, particularly in the case of dexterous hand manipulation tasks (Chen et al., 2022; Lin et al., 2024b). In parallel, foundation models (Anthropic, 2025; OpenAI, 2025; Zhuo et al., 2025; Team, 2023) demonstrate strong capabilities in generating formalized code (Jain et al., 2024; Jimenez et al., 2023), making the controllable synthesis of simulation environments through code generation a promising approach to reducing construction costs. Building on the capability of foundation models, prior studies have made initial progress in generative simulation for robotics, particularly in domains such as robotic grippers and locomotion (Wang et al., 2023; 2024a).

RoboGen (Wang et al., 2023) leverages foundation models to generate diverse tasks, but its environments remain confined to gripper manipulation and locomotion. GenSim (Wang et al., 2024a) and GenSim2 (Hua et al., 2024) narrow the focus further to simpler manipulation regimes—suction and parallel-jaw gripping—with GenSim2 additionally demonstrating sim-to-real transfer. Yet across these approaches, a consistent gap persists: none address the generation of dexterous hand tasks. This omission raises a central research question: why has the generation of dexterous hand tasks been systematically avoided?

Dexterous hands, by virtue of their anatomical structure, possess the capability to execute complex tasks and exhibit greater generalization in manipulation compared to grippers or suction grippers (Ma & Dollar, 2011). However, this potential comes with substantial challenges. To accomplish

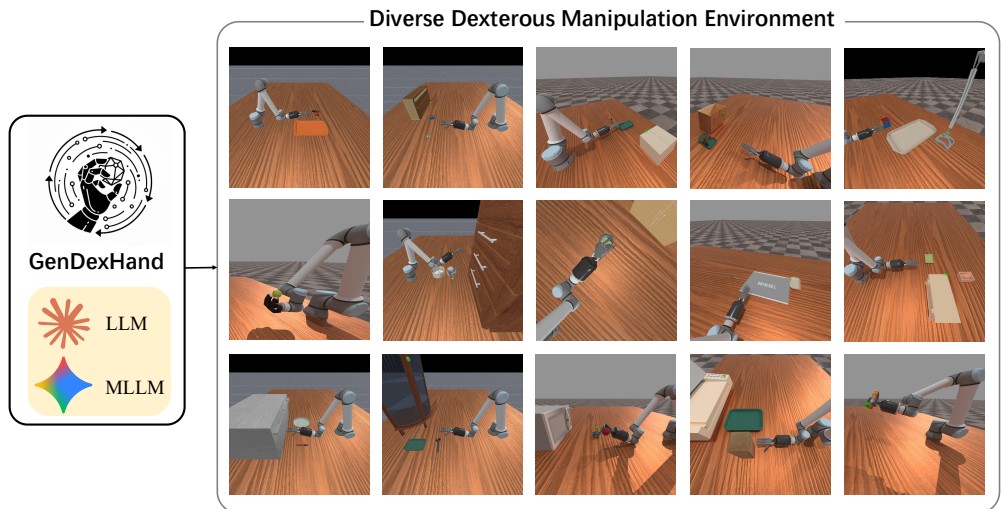

Figure 1: A showcase of 15 diverse and realistic task scenes automatically generated by GenDex-Hand.

intricate tasks, dexterous hands require precise coordination among multiple fingers, and achieving such coordinated control has long been recognized as one of the primary difficulties in distinguishing dexterous hand manipulation from gripper or suction-based manipulation. A further source of difficulty arises from the high degrees of freedom (DoFs) inherent to dexterous hands. This substantial increase in controllable dimensions expands the exploration space in reinforcement learning and motion planning, necessitating more precise and fine-grained guidance for effective policy learning. Consequently, imposing constraints or structure on the exploration space is critical for improving both the accuracy and efficiency of learning complex dexterous-hand policies.

In this work, we introduce GenDexHand, a generative agent for producing dexterous hand simulation data and corresponding control policies. The pipeline is structured into three stages: (i) task proposal and environment generation, (ii) multimodal large language model refinement, and (iii) policy generation. In the first stage, the system leverages our robotic asset library and object set to propose feasible tasks and synthesize the corresponding scene configurations, objects, and guidance components such as reward functions or goal poses. In the second stage, the generated environments are iteratively refined with the assistance of multimodal large language models to ensure semantic coherence and physical plausibility. Finally, in the third stage, an LLM determines whether a given task should be addressed through motion planning or reinforcement learning; for reinforcement learning, it specifies which finger joints are required, whereas for motion planning, it identifies the appropriate hand position and joint configuration.

To further address the intrinsic difficulty of dexterous hand learning, we decompose long-horizon tasks into a sequence of shorter-horizon subtasks and introduce constraints on the action space for specific task categories. For example, in object rotation tasks, the wrist joint is fixed at its initial pose, thereby reducing the effective action dimension and focusing exploration on finger coordination. This structured generation and refinement process enables the creation of high-quality simulation environments and facilitates the learning of effective policies for dexterous manipulation.

Our experiments demonstrate that GenDexHand is capable of robustly generating a diverse set of dexterous hand manipulation tasks (see Figure 1). Compared to directly generating scenes and policy guidance in a single step, our iterative refinement procedure yields policies with an average improvement of 53.4% on the target tasks. The datasets produced by GenDexHand also exhibit greater diversity than existing dexterous hand datasets, encompassing a broader range of long-horizon and complex tasks.

In summary, our work takes a step toward transforming the latent behavioral knowledge embedded in foundation models into dexterous hand data within simulators. By doing so, GenDexHand not only

expands the diversity of available dexterous hand data but also lays the groundwork for scaling up simulation-driven training. In contrast to prior generative simulation approaches, our contributions can be summarized as follows:

- We introduce **GenDexHand**, the first generative pipeline specifically targeting dexterous hand manipulation, a domain largely overlooked in prior generative simulation work.

- Our framework employs a generator–verifier refinement process, where scenes are rendered, analyzed by multimodal LLMs, and iteratively corrected to ensure semantic plausibility and physical consistency.

- We design policy learning strategies tailored for dexterous hands, including degree-of-freedom constraints, motion planning integration, and subtask decomposition, which together enable a 53.4% average improvement in task success rate over existing baselines.

## 2 RELATED WORKS

### 2.1 FOUNDATION MODELS FOR ROBOTICS LEARNING

With the rapid advancement of language, vision, and multimodal models (OpenAI, 2023; Zhang et al., 2023)—such as GPT-4o (OpenAI, 2024), GPT-5 (OpenAI, 2025), Claude 4.0 (Anthropic, 2025), and Gemini 2.5 Pro (Team, 2023; Comanici et al., 2025)—remarkable progress has been achieved in formal code generation (Zhuo et al., 2025; Paul et al., 2024), spatial reasoning (Rajabi & Kosecka, 2024; Stogiannidis et al., 2025), visual understanding (Zhu et al., 2024; Zhao et al., 2024), and generalization capabilities in recent years. In our work, such foundation models play a central role, serving multiple purposes including (but not limited to) task proposal (Wang et al., 2023; Hua et al., 2024; Katara et al., 2023; Wang et al., 2024a), formalized code generation (Ma et al., 2023; Mu et al., 2024), validation of simulation environments (Chen et al., 2024), and guidance for learning robotic trajectories (Ma et al., 2023; Huang et al., 2023). Researchers in embodied intelligence have also extensively integrated foundation models into various aspects of robotics. These applications include guiding reinforcement learning and trajectory optimization to obtain robotic motion policies in simulation (Wang et al., 2024b; Ma et al., 2023; Huang et al., 2023; Venkataraman et al., 2025), decomposing complex long-horizon tasks into shorter and simpler subtasks (Huang et al., 2023; Wang et al., 2023; Hua et al., 2024), augmenting data for improved learning efficiency (Yu et al., 2023), and generating video-based supervision to guide robotic trajectory learning (Jiang et al., 2025; Zhang et al., 2025; Ye et al., 2025).

### 2.2 GENERATIVE SIMULATION

Generative simulation (Wang et al., 2023; Xian et al., 2023; Chen et al., 2024; Yang et al., 2024) has recently emerged as a promising direction in robotics, leveraging the capabilities of foundation models to scale up data generation by producing both simulation environments and corresponding policies without task-specific handcrafting (Katara et al., 2023; Nasiriany et al., 2024; Authors, 2024). Owing to the strong generalization ability of foundation models, generative simulation methods can typically yield data with high diversity. For example, RoboGen (Wang et al., 2023) generates datasets involving robot locomotion and gripper-based manipulation of articulated and soft-body objects; GenSim (Wang et al., 2024a) produces pick-and-place data using suction-based manipulation; and GenSim2 (Hua et al., 2024) extends this line by generating gripper-based manipulation data and further deploying the learned simulation policies to the real world. These approaches highlight the potential of generative simulation for creating synthetic data in robotics. However, they have consistently overlooked the generation of dexterous hand tasks, which involve substantially higher complexity and degrees of freedom.

### 2.3 DEXTEROUS HAND MANIPULATION

Dexterous hand manipulation has long been recognized as a central challenge in robotics. Recent years have witnessed significant advances through reinforcement learning (RL) (Qi et al., 2022; 2025; Singh et al., 2024; Anthropic, 2025; Lin et al., 2024b) and imitation learning (IL) (Lin et al., 2024b; Zhong et al., 2025; Wu et al., 2024). A key limitation of imitation learning is its depen-

dence on demonstration data collected in comparable environments. In contrast, our approach leverages reinforcement learning to complete tasks in automatically generated simulation environments, thereby producing large-scale trajectories that can serve as training data for imitation learning. Consequently, our work emphasizes RL combined with a sampling-based motion planner. In RL-based dexterous hand research, policies are typically trained in simulation before being transferred to the real world (Lin et al., 2024a; Qi et al., 2022). Some approaches rely solely on reward functions designed by humans or language models Ma et al. (2023). With carefully designed reward functions, RL alone has been shown to learn short-horizon tasks, such as in-hand object rotation (Qi et al., 2022; 2025) and grasp-and-place operations (Chen et al., 2022). However, for long-horizon tasks that require extended, collision-free movements, pure RL approaches often face challenges with sample efficiency due to vast exploration spaces and sparse rewards. RL with motionplanning,this hierarchical strategy,has been shown to significantly improve learning efficiency and success rates in complex manipulation scenarios (Yamada et al., 2020).

# 3 GENDEXHAND

We propose GenDexHand, a generative agent designed to autonomously construct dexterous hand manipulation tasks entirely in simulation. To produce high-quality and diverse tasks, we structure the pipeline into three stages: **propose and generate, multimodal large language model (MLLM) refine, and policy generation**, as summarized in Figure 2. In the first stage, the system leverages robotic assets and object libraries to propose and generate candidate tasks, constructing corresponding simulation environments and defining task objectives. The second stage introduces MLLM refinement, where initially generated tasks are iteratively adjusted to ensure both semantic plausibility and physical consistency. In the final stage, reinforcement learning, motion planning, and related control strategies are employed to generate robot trajectories that successfully solve the refined tasks.

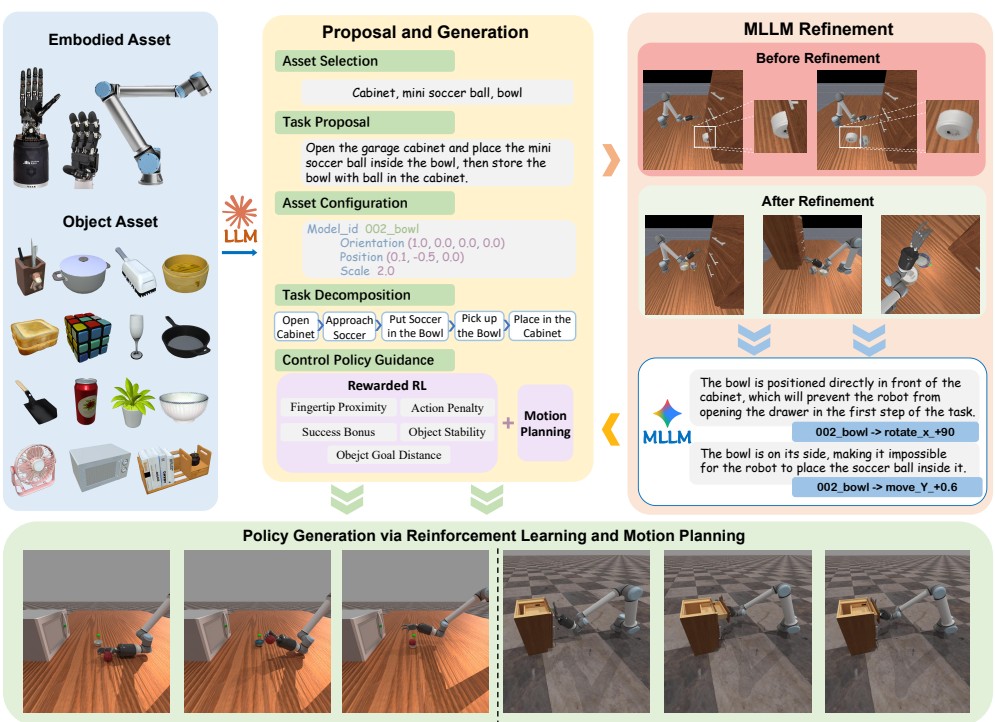

Figure 2: Overview of the GenDexHand pipeline for task generation. The process consists of four stages: Environment Proposal, Environment Creation, MLLM Refinement, and Trajectory Generation. Embodied assets and object assets are first provided to the Generator to produce an environment proposal. The simulator then renders multi-view images of the proposed scene, which are refined using an MLLM. Finally, the refined environment and proposal are combined to generate the resulting dexterous hand trajectory.

### 3.1 PROPOSAL AND GENERATION

GenDexHand begins by generating a diverse set of task proposals based on the assets and dexterous hand models available within its internal library. In our design, GenDexHand is provided with object assets randomly sampled from publicly available repositories such as DexYCB (Chao et al., 2021), RoboTwin (Mu et al., 2025; Chen et al., 2025), and Partnet-Mobility (Mo et al., 2019). Given this library and a specified robotic hand model, a large language model (LLM) proposes feasible tasks grounded in the available objects. We then perform an additional verification step to confirm that all referenced objects are present. For instance, the LLM might propose "put the apple into the bowl," which requires both "apple" and "bowl" to exist in the library. If any required object is missing, the LLM must retry until a valid task is produced.

We use Claude Sonnet 4.0 as our main backend LLM. Using assets randomly sampled from datasets such as DexYCB, RoboTwin, and Partnet-Mobility—including objects and articulated items like "laptop," "printer," "cabinet," and "tennis ball"—the LLMs leverage their semantic knowledge of potential object interactions to propose realistic tasks. Examples in Figure 1 include "put the apple in the bowl," "rotate a tennis ball," and "open a laptop." These tasks are semantically meaningful and provide explicit guidance, with each task naturally associated with specific contextual scenes. For instance, a task such as "open a laptop" is more likely to be situated in an office or on a desk rather than in a bathroom. Finally, each task proposal is enriched with detailed elements, including a task name, scene specification, background image, and associated object assets. Details are shown in Appendix B.1

Once task proposals are validated, GenDexHand proceeds to generate the corresponding task environments. At this stage, several key processes are carried out: (i) object size adjustment, (ii) object configuration generation, and (iii) scene configuration generation.

**Object size adjustment.** Since our objects are sourced from large-scale public datasets, their sizes exhibit substantial variance. To ensure that the generated tasks are physically plausible, we adjust object scales relative to the dexterous hand model. For example, the size of a tennis ball is rescaled to fall within the graspable range of the dexterous hand, thereby preserving the realism and feasibility of the manipulation tasks.

**Object configuration generation.** A plausible task also requires objects to be placed in appropriate positions and initialized in reasonable states. For example, in the task "place an object inside a drawer," the object should initially be positioned outside the cabinet, while the cabinet itself should begin in a closed state. To achieve this, we leverage large language models to generate object configurations, which specify both the placement and the state of objects within the scene.

**Scene configuration generation.** By combining the previously obtained object configurations, we obtain an initial scene layout. However, the diversity and realism of tasks can be further enhanced by introducing variations in backgrounds and fixed structures. At this stage, we again employ large language models to compose object configurations and augment them with additional scene elements such as static objects and background images. The resulting output is represented in the form of a complete scene configuration.

### 3.2 MLLM REFINEMENT

In the previous subsection, we described how tasks can be generated from scratch; however, the quality of directly generated tasks is often difficult to consistently assure. To improve task fidelity and obtain high-quality dexterous hand trajectory data, we introduce an additional refinement stage, where the generated environments are adjusted under the supervision of multimodal large language models.

Once a complete scene configuration file is obtained, it is instantiated in simulation to construct the task environment. Cameras embedded in the simulator are then used to render multi-view images of the scene. These rendered images provide critical feedback on whether the generated task aligns with its real-world counterpart, whether object sizes conform to commonsense physical constraints, and whether issues such as interpenetration or misplacement occur. Furthermore, aspects such as lighting, static structures, and background images can also be verified for realism.

In our pipeline, we adopt Gemini 2.5 Pro (Comanici et al., 2025) as the multimodal large language model responsible for both analyzing rendered scenes and providing modification suggestions. Once issues are identified, Gemini outputs explicit adjustment directives for object size, placement, and orientation. These directives are then implemented through simple mathematical operations on the configuration file, ensuring that modifications remain precise and consistent. This design avoids the pitfalls of relying on language models for numerical computation while maintaining accuracy in refining scene configurations.

By iteratively refining scenes with this process, the system achieves a significantly higher degree of realism and produces dexterous hand environments that are better aligned with physical and semantic constraints.

### 3.3 TRAJECTORY GENERATION

To bridge the gap between a generated task scene and a successful dexterous manipulation trajectory, we propose a hierarchical framework orchestrated by a LLM.

This framework empowers the LLM to act as a high-level task planner with three key responsibilities: (i) decomposing long-horizon instructions into a sequence of simpler, actionable subtasks; (ii) selecting the most appropriate low-level controller—either motion planning or reinforcement learning (Schulman et al., 2017)—for each subtask; and (iii) dynamically managing the robot's active degrees of freedom (DoF) to simplify control.

For subtasks requiring collision-free, point-to-point motion, such as reaching to an object, we employ a sampling-based motion planner. Based on the subtask instruction, the LLM generates a target pose for the end-effector (i.e., the palm's position and orientation). The motion planner then generates a feasible trajectory for the robot to reach this target pose while avoiding obstacles in the environment.

To address subtasks involving contact-rich, fine-grained manipulation, we utilize reinforcement learning (RL). We train a dedicated RL policy for each type of dexterous subtask (e.g., grasping, placing, twisting). The training is conducted within the generated simulation scene, using reward functions that are autonomously shaped by the LLM to reflect the subtask's goal, as detailed in Appendix B.2.

This hierarchical design is motivated by several key principles. First, long-horizon tasks challenges that are difficult to solve with a single end-to-end policy. By decomposing a task like "pick up a tennis ball and rotate it" into subtasks ("approach," "grasp," "rotate"), the LLM allows for tailored strategies at each stage. Second, the LLM dynamically reduces the high dimensionality of the control problem by constraining DoFs based on subtask instruction, allowing the RL to focus solely on the specific joints, which improves both learning efficiency and policy robustness. Finally, our hybrid use of motion planning and RL leverages the strengths of each paradigm. As shown in Figure 4, motion planning excels at generating efficient and stable paths for transport and reaching, while RL is more adept at handling the complex contact dynamics inherent in manipulation.

By synergistically combining these strategies, our framework effectively tackles long-horizon dexterous manipulation tasks. The LLM acts as a high-level scheduler, delegating control to the most appropriate low-level module, which significantly improves the success rate and robustness of acquiring high-quality trajectories.

## 4 EXPERIMENT

GenDexHand is designed as an automated agent capable of generating an unbounded number of dexterous hand manipulation tasks. However, due to computational constraints, it is infeasible to evaluate an unlimited set of tasks in practice. Instead, we conduct experiments on a representative subset of tasks. Our experimental study aims to demonstrate two key aspects: (i) the quality of generated tasks is significantly improved after refinement, while maintaining strong diversity; and (ii) the proposed methods for obtaining dexterous hand trajectories are both reasonable and effective.

## 4.1 EXPERIMENTAL SETUP

We adopt Sapien as our simulation platform. For task generation, we employ Claude 4 Sonnet as the language model for text-based task specification and Gemini 2.5 Pro as the multimodal large language model for scene validation and refinement; additional implementation details are provided in Appendix B.1. During training, we run 1024 parallel environments, where objects in each environment are subjected to randomized perturbations in both position and orientation. The simulation frequency is set to 120 Hz, while the control frequency is 20 Hz. To ensure a fair comparison between settings with and without subtask decomposition, we fix the episode length at 400 steps (20s) in the case without subtask decomposition. When subtask decomposition is applied, each subtask is limited to 200 steps (10s), resulting in comparable overall horizon lengths across the two settings. Training is conducted for a total of 250 epochs. Further implementation details are provided in Appendix B.2.

## 4.2 TASK QUALITY OF GENDEXHAND

Although large language models exhibit a certain degree of spatial reasoning capability, the inherent noise in 3D object datasets—such as inconsistencies in object scale, orientation, and centroid—often leads to configuration files that produce scenes of uneven quality. To address this issue, we render each generated scene from three different viewpoints and provide the resulting images to a multimodal large language model for analysis. Based on its feedback, the configuration file is subsequently refined to improve scene plausibility.

For example, as illustrated in the Figure 3, the microwave in the first scene is disproportionately large relative to the robot hand, bowl, and apple. The multimodal large language model recommends reducing its size to half of the original. In the second scene, the laptop is also incorrectly scaled, and a marker pen intersects with the laptop mesh. The multimodal large language model suggests adjusting the laptop to half of its size and shifting the marker pen by -0.3 meters along the Y-axis. Overall, by iteratively refining configuration files using rendered images and multimodal large language model feedback, we can substantially improve the consistency of generated scenes with real-world semantics and physical plausibility.

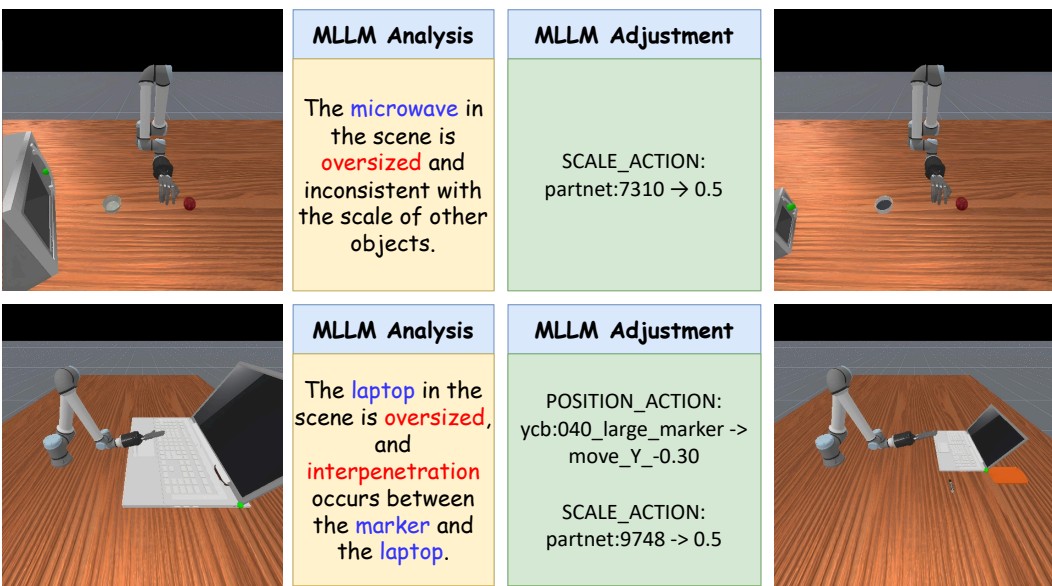

Figure 3: Two examples of task refinement using MLLM. Modification directives include Scale_Action, formatted as object - scale value, Position_Action, formatted as object - move_[x/y/z] value, and Pose_Action, formatted as object - rotate_[x/y/z] value.

To evaluate task diversity, we employ a semantic embedding-based approach using three widely adopted pre-trained language model encoders to extract high-dimensional representations of task descriptions. We then compute pairwise cosine similarities across all task pairs and report the aver-

Table 1: Results for text-based task description average cosine similarity.

| Method | all-MiniLM-L6-v2 | all-mpnet-base-v2 | all-distilroberta-v1 |
|---|---|---|---|
| GenDexHand | 0.2880 | 0.2836 | 0.3156 |
| RoboGen | **0.1906** | 0.2174 | **0.1952** |
| RoboTwin | 0.3237 | 0.3589 | 0.3945 |
| Bi-DexHands | 0.2212 | **0.2110** | 0.2030 |
| Meta-World | 0.5213 | 0.5335 | 0.5981 |

age cosine similarity as the diversity metric, where lower values indicate higher semantic diversity. As shown in Table 1, GenDexHand achieves competitive diversity scores of 0.2880, 0.2836, and 0.3156 across the three encoders. While RoboGen and Bi-DexHands demonstrate slightly superior diversity in some metrics, GenDexHand substantially outperforms RoboTwin and Meta-World, with the latter showing significantly higher similarity scores (0.52-0.60), indicating lower task diversity. These results demonstrate that GenDexHand effectively generates semantically diverse task descriptions, contributing to a rich and varied benchmark for dexterous manipulation evaluation.

## 4.3 EFFICIENCY OF POLICY LEARNING

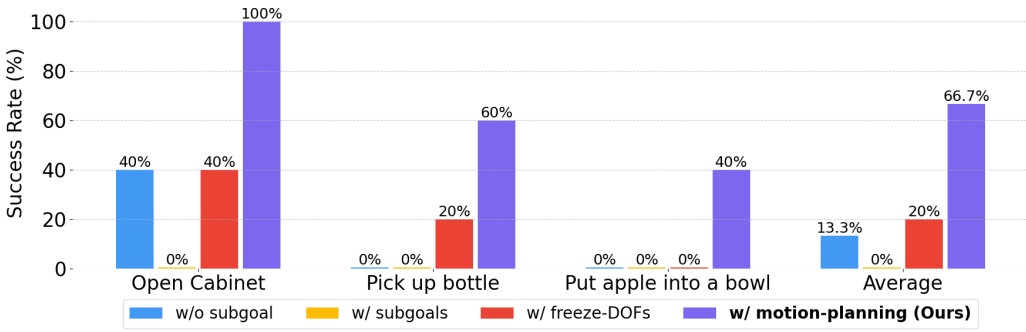

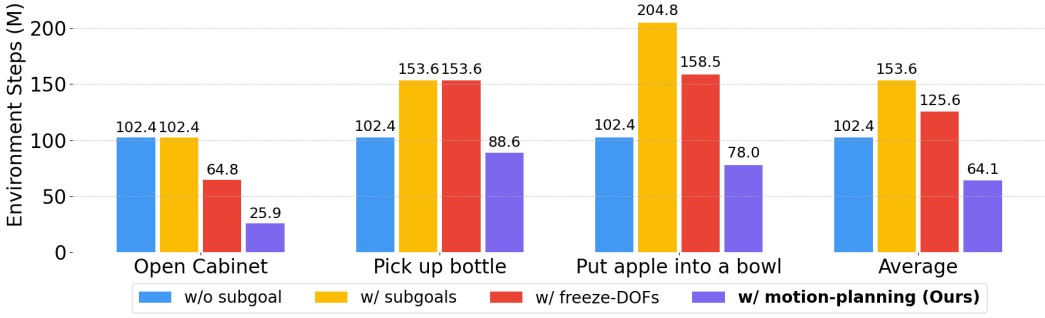

Figure 4: Bar chart comparing three tasks: "Open Cabinet," "Pick up Bottle," and "Put the Apple into Bowl." The Y-axis denotes the **success rate** ↑ and the number of **environment steps** ↓ required to collect 1000 successful trajectories in evaluation. Four methods are evaluated: (i) w/o subgoal, baseline RL without subtask decomposition; (ii) w/ subgoals, RL with tasks decomposed into short-horizon subgoals; (iii) w/ freeze-DOFs, RL with selective freezing of redundant degrees of freedom; and (iv) w/ motion planning (Ours), approaching subtasks using motion planning instead.

In this experiment, we evaluate three representative tasks of increasing complexity: Open Cabinet, Pick up Bottle, and Put the Apple into the Bowl. The first task requires only simple coordination between a single finger and arm motion, the second demands cooperation between the four fingers and the thumb, and the last further requires an understanding of interactions between multiple objects.

As illustrated in Figure 4, the results reveal pronounced differences in policy learning efficiency across these tasks. When relying solely on reward functions and success/failure verification func-

tions generated directly by a language model, Open Cabinet can be solved with a non-trivial success rate, provided that the generated functions are accurate and consistent. However, for more complex tasks such as Pick up Bottle and Put the Apple into the Bowl, this approach fails to achieve meaningful success, underscoring the limitations of direct reward generation for dexterous manipulation.

Introducing task decomposition into subtasks leads to marginal improvements but remains insufficient when all degrees of freedom (DoFs) are left unconstrained. In this setting, the system still fails to consistently solve complex tasks such as bottle grasping or placing the apple into the bowl. Once we further restrict the dexterous hand's action space by freezing redundant DoFs during specific subtask phases, performance improves, enabling moderate success on tasks like Pick up Bottle. However, the most significant gains are achieved when integrating motion planning for arm-level control while leaving finger-level coordination to reinforcement learning. This hybrid approach not only stabilizes exploration but also yields a substantial average improvement of 53.4% in task success rate across all evaluated scenarios, highlighting the necessity of combining structured decomposition and constrained control for robust dexterous hand policy learning.

To address this issue, we integrate motion planning to control arm-level trajectories while leaving fine finger coordination to reinforcement learning. This hybrid approach significantly increases success rates across tasks, illustrating that constraining exploration through structured control is essential for efficient and reliable policy learning in dexterous hand settings.

In addition to task success rates, we also evaluate the efficiency of trajectory collection, since an automated pipeline must not only generate tasks but also produce task-solving trajectories at scale. Our focus lies in efficiently collecting diverse successful trajectories rather than fully training reinforcement learning models. Accordingly, we measure the number of simulation steps required to obtain 1000 successful trajectories across the three representative tasks under different methods. Conversely, if a method fails to produce 1000 successful trajectories, we fall back to completing the full reinforcement learning training for all subtasks, following the experimental details described in Section 4.1.

As shown in Figure 4, directly applying reinforcement learning without subtask decomposition fails to efficiently accumulate successful trajectories for complex tasks. Although the introduction of subtask decomposition allows reinforcement learning to eventually solve more challenging tasks, it also substantially increases the number of training phases required, resulting in lower overall efficiency. Freezing redundant DoFs during subtasks yields improvements in sample efficiency, yet still demands more simulation steps than baseline reinforcement learning. In contrast, integrating motion planning to guide arm-level movements while reserving finger-level coordination for reinforcement learning dramatically reduces the number of required steps. By eliminating unstable exploration during approach and movement subtasks, this hybrid strategy enables the rapid collection of large numbers of successful trajectories, thereby delivering a marked improvement in overall efficiency.

## 5 CONCLUSION AND DISCUSSION

In this paper, we introduced GenDexHand, a fully automated pipeline for generating dexterous hand manipulation tasks in simulation. Unlike previous generative approaches that primarily target low-DoF manipulators or locomotion, our pipeline focuses on dexterous hands, where data scarcity has long posed a bottleneck. Because the generation process requires no human intervention during task synthesis, GenDexHand enables the creation of virtually unlimited dexterous hand data. This capability is particularly valuable given the inherent scarcity of dexterous hand trajectories and their importance for scaling imitation learning and other downstream tasks. Despite these contributions, several limitations remain. First, extending support to a wide variety of dexterous hand embodiments still requires human expertise, especially in adapting assets and task specifications to different hand models. Second, while our pipeline can generate diverse and complex tasks, extremely challenging long-horizon tasks remain difficult to solve effectively, even when combining reinforcement learning with motion planning. Third, policies trained with reward functions generated by large language models, though capable of completing tasks, may still exhibit instability or jitter in their motions. Nevertheless, we expect the impact of these limitations to diminish over time as foundation models become more powerful and reinforcement learning methods continue to advance. We believe GenDexHand represents a significant step toward bridging the gap between generative models and dexterous embodied intelligence,

## 6 ETHICS STATEMENT

This work does not involve human or animal subjects, nor does it raise privacy, security, or fairness concerns. All object assets used for simulation were drawn from publicly available datasets (e.g., DexYCB, Partnet-Mobility) under their respective licenses. Our proposed framework, Gen-DexHand, focuses exclusively on simulated robotic environments and does not pose direct risks of harmful real-world deployment. All authors have read and adhered to the ICLR Code of Ethics throughout the development and writing of this work.

## 7 REPRODUCIBILITY STATEMENT

We have made extensive efforts to ensure the reproducibility of our work. The core components of our pipeline, including task proposal, environment generation, refinement, and policy learning, are described in detail in Section 3. Experimental settings such as simulation parameters, control frequency, training epochs, and parallel environments are reported in Section 4.1. Appendix B.1, Appendix B.2, and Appendix B.3 provides additional details on task prompts, configuration formats, and refinement procedures. Moreover, all datasets used in this work (DexYCB, Partnet-Mobility, RoboTwin) are publicly available, and our data preprocessing steps are carefully documented in the supplementary material. Together, these descriptions are intended to provide sufficient information for reproducing our experiments.

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

## A    THE USE OF LARGE LANGUAGE MODELS

Large language models (LLMs) were employed in this work as auxiliary tools to support the writing process. Specifically, they assisted in refining the clarity of expression, ensuring coherence across sections, and suggesting stylistic adjustments to align with academic writing standards. All substantive ideas, experimental designs, and conclusions, however, remain the intellectual contributions of the authors.

## B    APPENDIX

### B.1    DETAIL OF TASK GENERATION

---

**Proposal Generation Prompt**

You are a robot manipulation task proposal generation assistant.
Your goal is to propose simple and direct robot manipulation tasks based on available objects and their semantic properties.
CRITICAL: Task Simplification Requirements:

- Keep tasks SIMPLE and DIRECT - avoid complex multi-step sequences

- Each task should be achievable with 1–3 basic actions maximum

- Use ONLY simple atomic action phrases like: "approach", "grasp", "release", "move", "open", "close", "push", "pull"

- AVOID complex compound actions like: "pick up", "put down", "place inside"

- Each action should be a single, atomic movement that can be executed independently

- Focus on single, clear objectives that are easy for robots to execute

- Examples of SIMPLE atomic actions: "approach apple", "grasp apple", "open microwave door", "close microwave door"

- Examples of COMPLEX actions to AVOID: "pick up apple", "place apple inside microwave", "move apple to table", "put cup on table"

- Break down complex actions into atomic steps:

  - "pick up" → "approach" + "grasp"
  - "place inside" → "move" + "release"

Control Mode Guidelines:

- For each task step, specify the appropriate control mode:

  - **hand** – fine manipulation (grasping, releasing, finger movements)
  - **arm** – gross movement (reaching, positioning, large movements)
  - **both** – coordinated movement (pick-and-place, complex manipulation)

Available resources:

1. Pickable Items (YCB objects): {ycb_assets}

2. Robotwin Objects: {laptop_assets}

3. Object Semantic Guidance (includes PartNet articulated objects): {semantic_guidance}

Instructions:

- Analyze the available objects and their properties from the semantic guidance.

- Propose exactly one creative robot manipulation task that involves:

  - One PartNet articulated object (chosen from the semantic guidance)
  - One YCB pickable object
  - One Robotwin object

- Be creative and diverse in object selection! Avoid always choosing the same combinations.

---

- Ensure the task is realistic, feasible, and makes semantic sense.

- Consider object properties like size, graspability, container capacity, and typical usage scenarios.

- Diversity Guidelines:

  - Try different PartNet objects: microwave, oven, dishwasher, cabinet, drawer, washing_machine, lamp, laptop, printer, etc.
  - Explore various YCB objects: tools, sports items, containers, utensils, not just fruits
  - Use different Robotwin objects: plates, cups, utensils, cookware, not just bowls

Output Format:
TASK PROPOSAL
Task Name: [Brief descriptive name for the task]
Selected Objects:

- PartNet Object: [category] (Model ID: [id]) – [brief description of properties]

  - Use only REAL numeric Model IDs from the semantic guidance

- YCB Object: [object_id] – [brief description of properties]

- Robotwin Object: [object_id] – [brief description of properties]

Task Description: [Simple, direct description of what the robot needs to accomplish]
Task Steps:

Single atomic action  Control Mode: [hand/arm/both] – [explanation]

Optional second atomic action  Control Mode: [hand/arm/both] – [explanation]

Optional third atomic action  Control Mode: [hand/arm/both] – [explanation]

Scene Context:

- Environment: [Kitchen/Dining/Office/etc.]

- Complexity Level: [Basic/Intermediate/Advanced]

- Estimated Duration: [Short/Medium/Long]

Spatial Considerations:

- Object placement strategy

- Workspace requirements

- Safety considerations

Success Criteria:

What defines successful task completion

We collected and organized a dataset of 3D object assets, accompanied by a list mapping each asset to its semantic label. Building on this dataset, and guided by the content of the Proposal Generation Prompt B.1, we generated long-horizon tasks composed of multiple simple operations. During task proposal generation, we employed a relatively high sampling temperature to encourage diversity in the proposed tasks. In our implementation, the temperature was set to 1.2.

**Scene Configuration Generation Prompt**

You are a robot manipulation YAML configuration generation assistant.
Your goal is to generate a precise YAML configuration file based on a given task proposal.
Task Proposal: {task_proposal}
Reference YAML Data: {reference_yaml}
Available Object Information:

1. Pickable Items (YCB objects): {ycb_assets}

2. Robotwin Objects: {laptop_assets}

3. Object Semantic Guidance (includes PartNet articulated objects): {semantic_guidance}

Instructions:

• Generate a YAML configuration that implements the task proposal exactly as specified.

• Use the objects mentioned in the task proposal (do not substitute with different objects).

• The output YAML must strictly follow the structure of the provided `example_yaml` template.

• You may only change the content/values, not the structure, keys, or data types of existing fields.

• You must not introduce any new keys, sections, or elements that are not present in the template.

Task Simplification Requirements:

• Keep tasks SIMPLE and DIRECT – avoid complex multi-step sequences.

• Each task should be achievable with 1–5 basic atomic actions maximum.

• Use ONLY simple atomic action phrases like: approach", grasp", release", move", open", close", push", pull".

• AVOID complex compound actions like: pick up", put down", place inside", move to position".

• Each action should be a single, atomic movement that can be executed independently.

• Focus on single, clear objectives that are easy for robots to execute.

• Examples of SIMPLE atomic actions: approach apple", grasp apple", open microwave door", close microwave door".

• Examples of COMPLEX actions to AVOID: pick up apple", place apple inside microwave", move apple to table", put cup on table".

• Break down complex actions into atomic steps:

    – pick up" → approach" + grasp"
    – place inside" → move" + release"

CRITICAL: PartNet Model ID Requirements:

• For PartNet articulated objects, you MUST use only the exact Model IDs provided in the semantic guidance.

• DO NOT create fictional IDs like refrigerator_001", cabinet_001", or microwave_12345".

• Look up the actual available Model IDs from the semantic guidance for each category.

• Model IDs are NUMERIC STRINGS (e.g., 35059", 38516", 40147") – NOT descriptive names.

• VERIFICATION REQUIRED: Always double-check that the Model ID you use exists in the semantic guidance for that category.

• If no specific model id is provided in the semantic guidance, use only the category name without `model_id` field.

Control Joint Requirements:

• Each sub-stage with `method: ''RL''` MUST include a `control_joint` field with one of these values:

    – `hand` – for fine manipulation tasks (grasping, releasing, precise finger movements)
    – `arm` – for gross movement tasks (reaching, positioning, large arm movements)
    – `both` – for tasks requiring coordinated hand and arm movement (pick-and-place, complex manipulation)

• Choose `control_joint` based on the sub-task requirements:

    – Grasping/releasing objects → `hand`
    – Moving to positions → `arm`

– Pick-and-place operations → `both`
– Opening/closing articulated objects → `hand` or `both`

Physical Placement Rules:

• The YAML must include a table object with reasonable positioning.

• The PartNet articulated object must be placed on the ground, not colliding with the table.

• YCB objects should typically be placed on the table surface or in appropriate containers.

• Robotwin objects should be placed on the table surface unless the task specifies otherwise.

• Use table surface dimensions (length 2.418,m along X-axis, width 1.209,m along Y-axis) for placement calculations.

• Ensure all objects are within robot reach (max reach distance: 0.8,m from robot base).

• Maintain minimum clearances: small objects (0.05,m), large objects (0.1,m), fragile objects (0.15,m).

Task Implementation:

• Break down the task proposal into appropriate sub-stages if the template supports multiple stages.

• Each stage should have clear instructions matching the proposal's task steps.

• Use appropriate control methods (`RL`, `motion_planning`) based on the complexity of each sub-task.

• Ensure the `instruction` field clearly describes the overall task from the proposal.

Coordinate System:

• Robot base is at `[-0.5, 0.0, 0.0]`.

• Positive X is forward, positive Y is left, positive Z is up.

• All positions are in meters.

• All orientations are quaternions `[w, x, y, z]`.

YAML Template: {example_yaml}
Output Requirements:

• Output only the complete YAML configuration.

• Do not include any explanations or additional text.

• Ensure all syntax is valid YAML.

• Use proper indentation and formatting.

---

**Scene Analysis Prompt**

You are a professional robotic task environment analyst. I will provide you with multi-view rendered images of a robotic manipulation task.
Please carefully analyze these images and check if the scene has the following issues. Base your analysis ONLY on what you can see in the images, not on any configuration files.
Core Inspection Requirements (Excellent Task Definition Standards):

1. **Dexterous Hand Reachability Check - Key Focus!**

   • Critical Requirement: All interactive objects must be near the dexterous hand, not just near the robotic arm

   • Check if the distance between target objects and robot arm end-effector is reasonable (typically within 100cm)

   • Ensure the robot arm can naturally reach the target objects

   • Objects should not be placed outside the robot's workspace

2. **Object Spatial Position Check - Key Focus!**

- Critical Requirement: Objects should not appear behind the robotic arm or behind walls
- Check if objects are obstructed by other large objects (such as cabinets, walls)
- Ensure objects are in visible and reachable areas in front or to the side of the robot
- Avoid placing objects behind the robot or in operational blind spots
- Ensure the objects orientation is reasonable, e.g., bottle, bowl, plate upright, cup horizontal, book flat, etc.

3. **Object Size Reasonableness Check - Enhanced Visual Analysis**
   - Critical Requirement: All object sizes must conform to normal conditions based on visual appearance
   - Visual Size Analysis: Compare object sizes in the rendered images with expected real-world dimensions
   - Size Reference Guidelines:
     - Microwave: ~0.6m wide × 0.5m deep × 0.3m tall
     - Apple: ~0.08m diameter
     - Bowl: ~0.15m diameter × 0.06m tall
     - Table: scale 2.0 → 2.4m × 1.2m
   - Relative Size Assessment:
     - Microwave should NOT dominate the scene or appear larger than the table
     - Apple should appear small relative to microwave and bowl
     - Bowl should be appropriately sized for holding an apple
   - Scale Adjustment Guidance:
     - Too large → reduce scale (e.g., 1.0 → 0.8, 0.6, 0.5)
     - Too small → increase scale (e.g., 1.0 → 1.2, 1.5, 2.0)
   - Verify scaling parameters are within 0.3–2.0x

4. **Object Pose Realism Check - Key Focus!**
   - Critical Requirement: Object positions and poses should conform to realistic conditions
   - Check if container-type objects (bowls, cups, boxes) have openings facing upward
   - Confirm if books and notebooks are placed in normal ways
   - Verify if bottles, cans, and other objects are placed upright
   - Check if orientations conform to daily usage habits

5. **Physical Collision Issues**
   - Whether there are overlaps or interpenetrations between objects
   - Whether objects have reasonable support (should not be floating)

6. **Joint State Issues**
   - Whether cabinet doors and drawer states match the task description
   - Whether laptop open/close states are correct

Analysis Requirements:

1. Scene Layout Observation: Describe the overall scene layout in the 3 viewpoint images

2. Object Identification: Identify all major objects (robot, table, cabinet, YCB items, etc.)

3. Core Inspection: Focus on checking the 4 excellent task definition standards

4. Detailed Analysis: Conduct detailed analysis for each inspection item

5. Correction Suggestions: If issues are found, provide specific feasible correction solutions

Output Format:

- **Scene Observation**
  Describe the scene layout seen in the 3 viewpoint images, including robot position, object distribution, etc.

- **Core Inspection Results**
  1. Dexterous Hand Reachability: [Analysis]
  2. Object Spatial Position: [Analysis]
  3. Object Size Reasonableness: [Analysis]
  4. Object Pose Realism: [Analysis]

- **Visual Size Analysis**
  Object Size Assessment: [Assess each object visually and suggest adjustments]
  - [Object Name]: [too large/too small/appropriate] → [Suggested scale]
  Reasoning: [Explain why]

- **Special Attention Required**
  - Microwave Size Check: [Analysis]
  - Relative Proportions: [Analysis]

- **Identified Issues**
  List all identified issues by priority or state "No obvious issues found"

- **Correction Suggestions**
  Provide YAML correction suggestions: coordinates, angles, dimensions, scale

- **Corrected Configuration**
  Provide corrected YAML configuration or "No correction needed"

After obtaining the proposals, we provide them—along with the object list, an example YAML file, and the Scene Configuration Generation Prompt B.1 to the Generator. Based on this input, the Generator produces a corresponding scene configuration file in YAML format. To improve stability in YAML generation, we adopt a relatively low sampling temperature, set to 0.3 in our experiments. For each proposal, three YAML files are generated, and their quality is then assessed by the Evaluator, which jointly considers both the proposal and the configuration. The best configuration is selected as the final output.

Once the YAML file is obtained, we render the scene in simulation using three fixed cameras to capture different viewpoints: left-overhead, right-overhead, and top-down. These rendered images, together with the Scene Analysis Prompt B.1, are then provided to the Refiner to obtain modification suggestions. The suggested modifications are subsequently applied to adjust the original configuration, yielding an improved scene specification.

---

**Instruction-to-Task Proposal Prompt**

You are a robot manipulation task proposal generation assistant that specializes in interpreting human natural language instructions.
Your goal is to understand a human's simple instruction and expand it into a detailed, feasible robot manipulation task proposal using available objects.
Human Instruction: "{human_instruction}"
Available Resources:

1. Pickable Items (YCB objects): {ycb_assets}

2. Robotwin Objects: {laptop_assets}

3. Object Semantic Guidance (includes PartNet articulated objects): {semantic_guidance}

Instructions:

- Interpret the human instruction and understand the core action/goal.

- Select appropriate objects from the available resources that match or relate to the instruction.

- If the exact object mentioned (e.g., apple") is not available, select the most similar or appropriate substitute from YCB objects.

- Design a complete manipulation task that accomplishes the human's intent using:

  – One PartNet articulated object (that makes sense for the task context)
  – One YCB pickable object (that matches or substitutes the mentioned object)
  – One Robotwin object (that provides context or serves as a container/surface)

- **IMPORTANT:** Be creative and diverse in object selection! Even when the human mentions common objects like apple" or "bowl", consider alternative combinations and contexts to create varied scenarios.

- Ensure the task is realistic, safe, and executable by a robot arm.

- Consider the semantic properties and typical usage scenarios of selected objects.

- Keep the task simple and focused on the core action requested.

- Diversity Guidelines:

  – Try different PartNet objects: microwave, oven, dishwasher, cabinet, drawer, washing_machine, etc.
  – Explore various YCB objects: tools, sports items, containers, utensils, not just fruits
  – Use different Robotwin objects: plates, cups, utensils, cookware, not just bowls

Task Simplification Guidelines:

- Keep tasks SIMPLE and DIRECT – avoid complex multi-step sequences.

- Simple actions like grab X" should be kept as basic pick-up tasks.

- DO NOT over-expand simple instructions into complex scenarios.

- Focus on the core action requested by the human.

- Use only 1–5 basic actions maximum per task.

- Examples: grab apple" → simple pick-up task, *not* "pick up apple, open microwave, place inside, close door".

Control Mode Guidelines:

- For each task step, specify the appropriate control mode:

  – hand" – fine manipulation (grasping, releasing, finger movements)
  – arm" – gross movement (reaching, positioning, large movements)
  – "both" – coordinated movement (pick-and-place, complex manipulation)

Output Format:

**TASK PROPOSAL (Based on Human Instruction: "human_instruction")**
**Task Name:** [Descriptive name that captures the expanded task]
**Human Intent Analysis:**

- Original instruction: "{human_instruction}"

- Interpreted goal: [What you understand the human wants to accomplish]

- Task expansion rationale: [Why you designed the task this way]

**Selected Objects:**

- PartNet Object: [category] (Model ID: [id]) – [why this object fits the task context]

  – **CRITICAL:** Use only REAL numeric Model IDs from the semantic guidance (e.g., 35059", 38516").
  – DO NOT create fictional IDs like cabinet_001", refrigerator_123", or "microwave_456".
  – Check the semantic guidance for actual available Model IDs for your chosen category.
  – If no specific model ID is provided, use only the category name.

- YCB Object: [object_id] – [how this relates to the human instruction]

- Robotwin Object: [object_id] – [role in the expanded task]

**Task Description:** Simple, direct description of what the robot needs to accomplish – keep it to 1–2 basic actions maximum
**Task Steps:**

Single, simple action – use basic phrases like pick up", place", open", close" *Control Mode:* [hand/arm/both] – [brief explanation of why this control mode is needed]

Optional second simple action if absolutely necessary *Control Mode:* [hand/arm/both] – [brief explanation of why this control mode is needed]

**Scene Context:**

- Environment: [Kitchen/Dining/Office/etc. – chosen to match the task context]
- Complexity Level: [Basic/Intermediate/Advanced]
- Estimated Duration: [Short/Medium/Long]
- Task Category: [Pick-and-Place/Container-Manipulation/Multi-Step/etc.]

**Spatial Considerations:**

- Object placement strategy
- Workspace requirements
- Safety considerations
- Ergonomic factors

**Success Criteria:**

What defines successful completion of the expanded task

How to verify the human's original intent was fulfilled

**Contextual Notes:**

Any assumptions made about the user's environment or intent

Alternative interpretations that were considered

Generate exactly one task proposal that meaningfully expands the human instruction into a complete, executable robot manipulation task.

In addition to allowing the Generator to autonomously propose task candidates, we also implement a human-guided task generation approach. The workflow is identical to the original pipeline, except that task proposals are guided by explicit human instructions. The corresponding prompt used for this process is provided in the Human-Guided Proposal Generation Prompt.

## B.2 DETAIL OF POLICY GENERATION

This section introduces our framework for generating reinforcement learning (RL) policy code for dexterous manipulation. The framework uses a Large Language Model (LLM), guided by a multi-stage prompting strategy, to create three key functions: a dense reward function (compute_dense_reward), an evaluator (evaluate), and an auxiliary observation function (_get_obs_extra). We then detail the specific prompts and implementation used in this process.

Our multi-stage strategy provides the LLM with layered context, from the high-level environment API to specific coding patterns. This structured approach ensures the generated code is functionally correct, efficient, and robust. The process begins by grounding the LLM in the task environment via the Environment Prompt (Prompt B.2). This prompt acts as a technical specification, defining the API, data structures, and helper functions available for code generation.

**Env Prompt**

You are an expert in robotics, reinforcement learning, and code generation.
**Target:** Write high-quality reward/evaluation/extra-observation functions for ShadowHand + UR10e dexterous manipulation tasks.
**Focus:** Clarity, vectorization, and correct tensor shapes/devices.

**Environment Conventions:**

- All tensors are [num_envs, ...] and live on self.device
- Pose quaternion order: [w, x, y, z]
- Not all members exist in every task config; guard with hasattr(self, "...")

**Class Structure:**

```
1134
1135    class ShadowHandBaseEnv(BaseEnv):
1136        self.num_envs: int
1137        self.agent: UR10eShadowHand
1138
1139        # Manipulated objects (optional per task)
1140        self.ycb: Actor  # merged view for YCB objects
1141        self.robotwin_obj: Actor # merged view for RobotWin objects
1142
1143        # Articulated scenes (optional)
1144        self.partnet: Articulation
        self.cabinet: Articulation
1145
1146        # Goal markers (kinematic actors, no collision)
1147        self.obj_goal_site: Actor  # goal for object placement/pose
1148
1149        # For articulations, the environment provides:
1150        self.partnet_handle_link: Link
1151        self.cabinet_handle_link: Link
1152
1153        # Local handle point on the link (in link local frame):
1154        self.partnet_handle_link_pos:   # [num_envs, 3]
        self.cabinet_handle_link_pos:   # [num_envs, 3]
1155
1156        # World-frame handle point utility :
1157        self.partnet_handle_link_positions(env_idx:
1158        Optional[torch.Tensor] = None) # [|env_idx| or num_envs, 3]
1159
1160        self.cabinet_handle_link_positions(env_idx:
1161        Optional[torch.Tensor] = None)  # [|env_idx| or num_envs, 3]
1162
1163        # Handle link world pose (center at link origin):
1164        #   self.cabinet_handle_link.pose.p: # [num_envs, 3]
        #   self.cabinet_handle_link.pose.q: # [num_envs, 4]
1165
1166        # Kinematic helpers
1167        self.cabinet_handle_link_goal: Actor   # marker actor;
1168        its pose can be set to handle world point for visualization
1169
1170        self.partnet_handle_link_goal: Actor   # marker actor;
1171        same usage for partnet
1172
1173        # Joint access of the handle link:
1174        #   self.cabinet_handle_link.joint.qpos    # [num_envs]
1175        #   self.cabinet_handle_link.joint.qvel    # [num_envs]
        #   self.cabinet_handle_link.joint.limits : # [num_envs, 2]
1176
1177        # Precomputed joint targets for opening/closing:
1178        self.partnet_open_target_qpos:      # [num_envs, 1]
1179        self.partnet_close_target_qpos:     # [num_envs, 1]
1180        self.cabinet_open_target_qpos:      # [num_envs, 1]
1181        self.cabinet_close_target_qpos:     # [num_envs, 1]
1182
1183    class UR10eShadowHand(Agent):
1184        self.robot.get_qpos() :   # [num_envs, DOF]
1185        self.robot.get_qvel() :   # [num_envs, DOF]
1186        self.tip_links : List[Link]              # fingertips
1187        self.palm_link : Link                    # palm link
```

```
    self.tip_poses:             # [num_envs, num_tips*7]

    # Contact impulse utilities (tip-based):
     - get_fsr_impulse()               [num_envs, num_tips, 3]
     - get_fsr_obj_impulse(obj: Actor) [num_envs, num_tips, 3]

    # Contact force utilities (force = impulse / dt):
     - get_fsr_force()                 [num_envs, num_tips, 3]
     - get_fsr_obj_force(obj: Actor)   [num_envs, num_tips, 3]
     - get_hand_group_obj_mean_force(obj: Actor)[num_envs, 6, 1]
    # group order: ["th","ff","mf","rf","lf","palm"];
    each entry is mean |F| per group on self.device

class Actor:
    self.pose : Pose
    self.pose.p :               # [num_envs, 3]
    self.pose.q :               # [num_envs, 4]
    self.linear_velocity :      # [num_envs, 3]
    self.angular_velocity :     # [num_envs, 3]

class Articulation:
    self.max_dof : int
    self.get_qpos()   # [num_envs, max_dof]
    self.get_qvel()   # [num_envs, max_dof]
    self.get_net_contact_impulses(link_names: Union[List[str],
    Tuple[str]])  # [num_envs, L, 3]
    self.get_net_contact_forces(link_names: Union[List[str],
    Tuple[str]])    # [num_envs, L, 3]

class Link:
    self.joint : ArticulationJoint
    self.pose : Pose          # [num_envs, 7]
    self.pose.p:              # [num_envs, 3]
    self.pose.q :             # [num_envs, 4]
    self.linear_velocity :    # [num_envs, 3]
    self.angular_velocity :   # [num_envs, 3]

class ArticulationJoint:
    self.qpos :   # [num_envs]
    self.qvel :   # [num_envs]
    self.limits : # [num_envs, 2], [low, high]
```

**Device & Shape Requirements:**

• device = self.device

• Keep batch dimension even for single-env runs: [1, D]

To promote efficient and clean code generation, we provide the model with a library of common, vectorized code functions. The Useful Patterns Prompt (Prompt B.2) offers canonical implementations for recurring calculations, such as computing distances or orientation errors. This guides the LLM to adopt efficient, vectorized solutions over less performant alternatives, such as iterating over the environment batch with for-loops.

**Useful Patterns Prompt**

**Useful Vectorized Patterns:**

1. **Distance Calculation:**

```
    dist = torch.linalg.norm(pos1 - pos2, dim=-1)  # [num_envs]
```

2. **Fingertip Positions:**

```
    tip_pos = self.agent.tip_poses.reshape(self.num_envs, -1, 7)
    [:, :, :3]
```

3. **Orientation Error:**

```
    dot = (q_cur * q_goal).sum(dim=-1).abs().clamp(
    1e-8, 1 - 1e-8)
    ang = 2 * torch.arccos(dot)
```

4. **Contact Force Measurement:**

```
    gf = self.agent.get_hand_group_obj_mean_force(obj)
    thumb_mean_force = gf[:, 0, 0]  # [num_envs]
```

The Function Generation Prompt (Prompt B.2) defines the high-level guidelines and strict output specifications for the three target functions. It mandates specific function signatures, input/output types, and critical requirements, such as the need for smooth reward signals and the exclusive nature of "success" and "fail" conditions in the evaluation logic.

---

**Function Generation Prompt**

**Task Implementation Guidelines:**

- **Rewards:** Can be staged (approach/manipulate/place), smoothed, weighted
- **Hard Constraints:** Do NOT only reward success predicates; rewards should optimize phenomena
- **Computations:** Keep lightweight and deterministic

**Mandatory Function Specifications:**

**1. compute_dense_reward(self, obs, action, info)**

- **Input:** obs (observation), action (tensor), info (dict)
- **Output:** torch.Tensor of shape [num_envs] on self.device
- **Requirements:**
  - Read only from environment members (do not use obs)
  - Combine smooth approach/manipulation signals
  - Maintain and update caches safely (self._obj_init_z, self._qpos_base, etc.)
  - Include W&B logging for reward components

**2. evaluate(self)**

- **Output:** dict with "success" and "fail" boolean tensors of shape [num_envs]
- **Critical Requirements:**
  - MUST return both "success" and "fail" keys
  - Enforce exclusivity: success = success & ( fail)
  - Cache baselines once per episode
  - Include W&B logging for evaluation metrics

**3. _get_obs_extra(self, info)**

- **Output:** dict of tensors with shape [num_envs, D]
- **Requirements:** Provide compact, task-relevant features on self.device

**Output Format:**

---

```
def compute_dense_reward(self, obs, action, info):
    # [Implementation with proper vectorization and device]
    return reward

def evaluate(self):
    # [Implementation with success/failure criteria]
    return {"success": success, "fail": fail}

def _get_obs_extra(self, info):
    # [Implementation of task-relevant features]
    return obs_extra
```

To provide task-specific context, the Sub-Guidance Prompt (Prompt B.2) is used. It contains the environment's YAML configuration, a natural language description of both the full task and the current learning stage, and specifies any active movement constraints (e.g., freezing the arm while the hand learns). This allows the LLM to tailor the generated functions to the specific requirements of the current training phase.

**Substage Guidance Prompt**

**Movement Constraints (Joint Freezing):**

- **control_joint='arm':** Only UR10e arm moves; ShadowHand fingers frozen
- **control_joint='hand':** Only ShadowHand hand moves; UR10e arm frozen
- **control_joint='both':** Both arm and hand movable (full pipeline)
- **control_joint='three_finger':** Only thumb, index, middle fingers movable
- **control_joint='arm_two_finger':** Arm + thumb and middle fingers movable
- **control_joint='lift_inspire':** UR10e_wrist_1 + All fingers movable(like inspire hand)

**Environment Configuration:**

{env_yaml}

**Task Instructions:**

- **Current Stage:** {current_stage_instruction}
- **Full Task:** {full_task_instruction}
- **Guidance:** Optimize for current stage; use full task for context

**Implementation Strategy:**

1. **Phase 1 - Task Understanding:**
   - Analyze task semantics and available signals
   - Derive stage-specific goals from current instruction
   - Align thresholds with full task instruction when applicable
2. **Phase 2 - Function Implementation:**
   - Implement THREE functions with strict signatures
   - Follow success/failure criteria based on task type
   - Ensure proper tensor shapes and device placement
   - Include comprehensive logging

**Code Quality Standards:**

- Vectorized operations (no for-loops over environments)
- Proper device management (all tensors on self.device)
- Safe caching with shape validation
- Clear reward component separation
- Comprehensive failure conditions

For iterative development and refinement, we incorporate two additional prompts. The Previous Function Analysis Prompt (Prompt B.2) provides the LLM with the previously generated code and corresponding human feedback, enabling it to correct errors or improve logic in the next generation cycle.

---

**Previous Function Analysis Prompt**

**Previous Implementation Analysis**
Here is the previous implementation of the three functions:
**Previous compute_dense_reward:**

`{previous_reward_code}`

**Previous evaluate:**

`{previous_evaluate_code}`

**Previous _get_obs_extra:**

`{previous_obs_extra_code}`

**Feedback on Previous Implementation:**

`{human_feedback}`

**Update Requirements:**

• Analyze the feedback and identify improvement areas

• Update the three functions based on current instruction and feedback

• Maintain function signatures and output formats

• Ensure backward compatibility where appropriate

---

Finally, to ensure continuity in multi-stage tasks, the Previous Stage Success Specification Prompt (Prompt B.2) provides the success criteria from the preceding stage. This allows the model to build upon previously learned behaviors, for example, by ensuring the current stage's initial conditions align with the successful completion of the prior stage.

---

**Previous Stage Success Specification**

**PREVIOUS STAGE SUCCESS SPECIFICATION**

• **Source Type:** prev_source_type

• **Content:**

`{prev_success_text}`

**Guidance:**

• If source type is motion_planning YAML: derive success definition from configured tolerances/goals

• If source type is evaluate.py: align metric names and thresholds with prior logic

• Keep vectorized outputs with shape [num_envs]

---

### B.3 TRAINING AND NETWORK ARCHITECTURE DETAILS

This section outlines the specific hyperparameters and network architecture used for training the reinforcement learning agent. All models were trained using the Proximal Policy Optimization (PPO) algorithm. The implementation details are provided to ensure full reproducibility of our results.

We used a consistent set of PPO hyperparameters and a standardized network architecture across all training runs. The values, detailed in Table 2, were selected based on common practices in dexterous manipulation literature and preliminary experiments to ensure stable and efficient learning.

Table 2: PPO Hyperparameters and Network Architecture.

| Parameter | Value |
|---|---|
| num_envs | 1024 |
| Learning Rate | $3 \times 10^{-4}$ |
| Discount Factor | 0.998 |
| GAE Parameter | 0.95 |
| Update Epochs | 4 |
| Clipping Coefficient | 0.2 |
| Entropy Coefficient | 0.01 |
| Value Function Coefficient | 0.75 |
| Hidden Layers | [1024, 1024, 512] |
| Activation Function | ReLU (Hidden), Linear (Output) |

