# OpenReview forum: "Generative Simulation for Dexterous Hands"
_ICLR.cc/2026/Conference — Submitted to ICLR 2026_

### Official Review · Reviewer_B3uT · 2025-10-20

**Soundness:** 3
**Presentation:** 3
**Contribution:** 3
**Rating:** 6
**Confidence:** 3

**Summary:**

The authors propose a generative simulation pipeline to produce robotic tasks and environments for dexterous manipulation. It consists of three stages: (i) task proposal and environment generation (using Claude Sonnet 4.0 and assets sampled from DexYCB, RoboTwin/Robotwin, PartNet-Mobility, (ii) multimodal large language model refinement (using Gemini Pro 2.5 to explicitly adjust object size, placement, andorientation), and (iii) policy generation (using motion planning + RL).

**Strengths:**

- The idea is interesting and new.
- The generative pipeline design seems solid.
- paper is well written and well structured. the examples given in the paper are easy to follow and the supplementary materials provide quite a lot details.

**Weaknesses:**

- My biggest concern is the efficiency and effectiveness of the generative method in terms of sim2real. The authors did not provide any sim2real experiments to evaluate the quality of their generated data.
- The experiments only use a small set of tasks.
- Figure 2 caption description is not consistent with the previous statement. It says the process consists of four stages, in which it counts environment proposal and generation as two separate stages, whereas in the introduction section, it claims the pipeline consists of three stages and counts environment proposal and generation as one stage.

**Questions:**

- Does the RL policy need to be retrained for every task generated? or the RL policy can be trained on one/several subtasks and generalize across tasks?
- How many hours does it take to generate X samples on Y device?

---

> ### Author Response · Authors · 2025-11-27
>
> > My biggest concern is the efficiency and effectiveness of the generative method in terms of sim2real. The authors did not provide any sim2real experiments to evaluate the quality of their generated data.
> - Thank you for highlighting this important point. As noted in our author response to all reviewers, where we report how policies trained on our generated data perform when deployed in real robot.
>
> > The experiments only use a small set of tasks.
> - Our framework is capable of generating a much larger variety of tasks—in total, 50 tasks were created. However, running all baselines across the entire task set would be prohibitively expensive. Therefore, we selected a representative subset that spans different manipulation patterns, difficulty levels, and object interaction types. These tasks provide a meaningful and balanced evaluation of the framework while keeping computational cost manageable.
>
> > Figure 2 caption description is not consistent with the previous statement. It says the process consists of four stages, in which it counts environment proposal and generation as two separate stages, whereas in the introduction section, it claims the pipeline consists of three stages and counts environment proposal and generation as one stage.
> - To maintain clarity and consistency throughout the manuscript, we will standardize the description to four stages across all sections. The caption of Figure 2 and the introduction will be updated accordingly in the revised version.
>
> > Does the RL policy need to be retrained for every task generated? or the RL policy can be trained on one/several subtasks and generalize across tasks?
> - In our current implementation, the RL policy is trained separately for each task, as different tasks often involve distinct object interactions, reward structures, and control strategies. This per-task training ensures that each generated task can be solved reliably.
> - That said, one of our broader goals is to enable imitation-learning-based policies trained on our generated dataset to generalize across tasks. Since the framework provides a diverse set of demonstrations spanning multiple skill families, we believe this opens the door to training a single policy that can transfer across tasks without retraining.
>
> > How many hours does it take to generate X samples on Y device?
> - For clarity, generating one complete task— including environment generation, trajectory synthesis, and data refinement—takes approximately 10 hours per task on an NVIDIA RTX 4090 GPU.

---

### Official Review · Reviewer_DQ93 · 2025-10-26

**Soundness:** 2
**Presentation:** 2
**Contribution:** 2
**Rating:** 2
**Confidence:** 4

**Summary:**

This paper proposes a generative simulation pipeline for dexterous hand manipulation. It has several stages—task proposal and environment generation, MLLM refinement, and policy generation, with designs including closed-loop MLLM-driven scene adjustment, subtask decomposition, and hybrid motion planning/reinforcement learning (RL) for policy training. Experiments show it can generate physically plausible tasks and achieves a 53.4% average improvement in task success rate compared to baselines.

**Strengths:**

1. The entire pipeline appears to be feasible.
2. The paper is well-structured and clearly written.

**Weaknesses:**

1. The work represents only an incremental improvement over existing gripper-based data generation approaches [1,2] and lacks novelty.

2. Most experiments in the paper do not actually require a dexterous hand — the tasks can largely be accomplished with a simple gripper, revealing a lack of truly dexterous manipulation tasks. Only a in-hand manipulation task truly needs dexterous hand, however this task don't need this pipeline actually, because a lot of previous works have done it well.

3. The provided video demonstrations are short and feature relatively simple tasks, lacking examples of complex or long-horizon manipulation.

4. The work lacks the results of imitation learning experiments trained on its data. The collected data needs to be validated for its effectiveness in training autonomous policies — otherwise, the dataset itself holds limited practical significance.

5. Due to the absence of real-world robot experiments, the authors fail to demonstrate the practical usefulness of the proposed pipeline for real dexterous robotic manipulation.

[1] GenSim2: Scaling Robot Data Generation with Multi-modal and Reasoning LLMs

[2] RoboTwin 2.0: A Scalable Data Generator and Benchmark with Strong Domain Randomization for Robust Bimanual Robotic Manipulation

**Questions:**

See Weakness.

---

> ### Author Response · Authors · 2025-11-27
>
> > The work represents only an incremental improvement over existing gripper-based data generation approaches [1,2] and lacks novelty.
> - GenSim2 and RoboTwin primarily focus on trajectory generation or imitation within pre-defined tasks, rather than automatically generating diverse tasks end-to-end. In contrast, our framework explicitly targets task generation, which requires reasoning about scene configuration, object affordances, and multi-stage goal structures. Moreover, extending task generation to dexterous hands introduces substantially higher complexity compared to gripper-based systems. Dexterous hands have significantly higher degrees of freedom, making even basic behaviors—such as stable grasping, coordinated finger motion, or in-hand manipulation—far more challenging. These capabilities demand both richer task specifications and more precise control strategies, which are not addressed by existing gripper-centric frameworks.
>
> > Most experiments in the paper do not actually require a dexterous hand — the tasks can largely be accomplished with a simple gripper, revealing a lack of truly dexterous manipulation tasks. Only a in-hand manipulation task truly needs dexterous hand, however this task don't need this pipeline actually, because a lot of previous works have done it well.
>
> - First, our benchmark does include in-hand manipulation tasks, which inherently require dexterous, multi-fingered control and cannot be solved by simple parallel-jaw grippers.
>
> - Second, while it is true that several tasks in our benchmark can be executed by a simple gripper, we argue that this does not diminish the value of generating data specifically for dexterous-hand manipulation. Training robust dexterous-hand policies requires a broad spectrum of demonstrations, including both:
> (1) tasks that cannot be completed by a gripper (e.g., in-hand rotation), and
> (2) “simpler” tasks that are still non-trivial for dexterous hands due to their high DOF, complex actuation, and coordination requirements.
>
> - Even grasping or placing with a dexterous hand is significantly more challenging than with a gripper. Therefore, generating diverse data—even for tasks that may appear simple—is essential for learning stable, generalizable dexterous-hand control. This pipeline fills an important gap by providing scalable task and data generation specifically tailored for dexterous manipulation, rather than assuming gripper-centric solutions transfer directly.
>
> > The provided video demonstrations are short and feature relatively simple tasks, lacking examples of complex or long-horizon manipulation.
>
> - We thank the reviewer for the comment and would like to clarify what “long-horizon” means in the context of dexterous manipulation. In our setting, a long-horizon task refers to a sequence of multiple goal-directed phases that require sustained control, continuous state tracking, and coordinated hand–object interaction (e.g., reach → grasp → transport → precise placement). Although tasks such as “pick and place the apple into a bowl” or “pick and place the drill on the plate” may appear simple at first glance, they in fact constitute multi-stage manipulation processes when executed with a high-DOF dexterous hand.
>
> > The work lacks the results of imitation learning experiments trained on its data. The collected data needs to be validated for its effectiveness in training autonomous policies — otherwise, the dataset itself holds limited practical significance.
> - We agree that demonstrating downstream policy learning is important. As noted in our author response to all reviewers, where we report how policies trained on our generated data perform when deployed in both simulation and real hardware.
>
> > Due to the absence of real-world robot experiments, the authors fail to demonstrate the practical usefulness of the proposed pipeline for real dexterous robotic manipulation.
> - As noted in our author response to all reviewers, where we report how policies trained on our generated data perform when deployed in real robot.

---

### Official Review · Reviewer_MgvZ · 2025-11-01

**Soundness:** 2
**Presentation:** 2
**Contribution:** 2
**Rating:** 2
**Confidence:** 4

**Summary:**

The authors present a GenDexHand, a simulation framework to generate tasks and data, tailored specifically for dextrous manipulation tasks. The use LLMs to generate tasks, MLLMs to refine them, and they benchmark various methods to generate data for these tasks. While the ideas are sound and experiments are promising, there are several missing details in the paper about the scope of the benchmark, and experiments are sparse. Please see comments below for additional details.

**Strengths:**

- the prompts to generate tasks and data are elaborate and well-thought out. they are clearly laid out in the appendix, making it transparent how the system works.
- using MMLMs to refine tasks is a practical idea, and the authors show specific examples how this is applied in practice to obtain more realistic tasks.

**Weaknesses:**

- The experiment to quantify task diversity via cosine similarities of the text embeddings is just one specific metric, but is not a holistic way to measure task diversity. For example, how many assets are incorporated? How many skill families are present, compared to other works? What is the distribution of the number of stages per task? How many environments are present? This information is missing in the current manuscript.
- The experiments only feature three tasks (figure 4), and the tasks are not very diverse (two are basic pick-place tasks). I presume there are more tasks that this framework can generate, but the main text does not mention the full scope of tasks.
- The paper presents a simulation framework with the goal of generating diverse data, presumably to build real-world robot agents, but there are no experiments or discussion about how to use the generated data for transfer to real world environments and tasks.

**Questions:**

- How many tasks in total are generated by this simulation framework? What are the skill families present?
- It's unclear why without subtask decomposition, the episode length is 400 steps, but with subtask decomposition the episode length is 200 steps. Is this a fair comparison?

---

> ### Author Response · Authors · 2025-11-27
>
> > The experiment to quantify task diversity via cosine similarities of the text embeddings is just one specific metric, but is not a holistic way to measure task diversity. For example, how many assets are incorporated? How many skill families are present, compared to other works? What is the distribution of the number of stages per task? How many environments are present? This information is missing in the current manuscript.
>
> 1. In total, our pipeline generated 50 distinct tasks, each with different objects, layouts, and manipulation requirements.
> 2. Object Asset Diversity.
>     Our pipeline incorporates a large and diverse set of manipulable assets from multiple well-established datasets, including:
>     - PartNet-Mobility: 1,284 articulated objects
>     - YCB: 78 everyday objects
>     - RoboTwin: 16 task-oriented objects
> 3. The 50 generated tasks encompass more than six distinct skill families, including: grasping, placing, opening articulated objects, in-hand rotation, closing articulated objects, and interact with articulated objects.
> 4. Distribution of number of stages
>     | **Stage Count** | 1 | 2 | 3 | 4  | 5  | 6 | 7 | 8 |
>     | --------------- | - | - | - | -- | -- | - | - | - |
>     | **# of Tasks**  | 2 | 3 | 9 | 16 | 10 | 3 | 2 | 5 |
>
>
>
> > The experiments only feature three tasks (figure 4), and the tasks are not very diverse (two are basic pick-place tasks). I presume there are more tasks that this framework can generate, but the main text does not mention the full scope of tasks.
>
> - Our framework is indeed capable of generating a significantly larger set of tasks. In total, we generated 50 tasks across different categories. However, due to space constraints and the cost of running all baselines on every task, we did not perform exhaustive comparisons across the entire task set. Instead, we selected a subset of tasks that are representative in terms of object interaction patterns, motion complexity, and horizon length. These tasks provide a balanced evaluation of the framework’s capabilities while keeping the experimental budget reasonable. We will clarify this in the revised version and include a more detailed description of the full task set in the appendix.
>
>
> > The paper presents a simulation framework with the goal of generating diverse data, presumably to build real-world robot agents, but there are no experiments or discussion about how to use the generated data for transfer to real world environments and tasks.
>
> - As noted in our author response to all reviewers, where we report how policies trained on our generated data perform when deployed in real robot.
>
> > How many tasks in total are generated by this simulation framework? What are the skill families present?
>
> - Using the current simulation framework, we generated a total of 50 tasks. These tasks cover a broad range of skill families, including grasping, placing, opening articulated objects, closing articulated objects, in-hand rotation, and general interaction with articulated objects. This diversity reflects the framework’s ability to support both basic manipulation skills and more complex, long-horizon dexterous behaviors. We will add a clearer description of the full task set in the revised manuscript and appendix for completeness.
>
> > It's unclear why without subtask decomposition, the episode length is 400 steps, but with subtask decomposition the episode length is 200 steps. Is this a fair comparison?
> - Thank you for raising this point. The non-decomposed (single-horizon) versions of the tasks generally require substantially more interaction steps to complete, as the policy must solve the entire long-horizon task without intermediate guidance. To ensure a fair comparison, we therefore allocate twice the episode horizon (400 steps) for the non-decomposed setting, whereas each decomposed subtask is assigned a 200-step horizon. This rationale is already discussed in the main text (Lines 331–334), and we will further clarify it in the revision to avoid ambiguity.

---

### Official Review · Reviewer_NbQh · 2025-11-01

**Soundness:** 2
**Presentation:** 2
**Contribution:** 2
**Rating:** 2
**Confidence:** 4

**Summary:**

This paper presents GenDexHand, which uses VLMs to automate environment and data generation in simulation for dexterous hand manipulation tasks. The main contributions are:
1. It focuses on dexterous hand manipulation.
2. It uses VLMs not only to create tasks but also to check and refine them.
3. It studies policy learning for the proposed tasks, including task decomposition and motion planning integration.
Experiments are done on several simulation tasks, and task diversity is reported.

**Strengths:**

The writing is clear and easy to follow.
It is good to explore using VLMs for dexterous task design and simulation setup.
It is also good to study reinforcement learning, frozen joints, and motion planning.

**Weaknesses:**

- The main weakness is that there are no real-world experiments. This means the paper cannot show if the simulation is actually useful in real applications, which makes it less convincing for manipulation research.
- Another weakness is that the idea of using VLMs for simulation setup is no longer new (unlike GenSim or RoboGen). Many researchers now question the value of VLM-generated simulations. Reviewers may expect a solid and practical simulation benchmark such as THOR or RoboCasa. The authors are encouraged to build a far more diverse and convincing VLM-based simulation benchmark that includes extensive real-world validation. For the tasks in this paper, setting them up manually would be easier and more controllable than using prompts.
- From the policy learning point of view, compared to recent studies, it is hard to say that subtask decomposition, motion planning, or freezing degrees of freedom are contributions. If the authors want to highlight policy learning, they should provide more insights, compare with state-of-the-art methods and include real-world demonstrations.
- The results in Table 1 are also not convincing, and more experiments are recommended.

**Questions:**

See weaknesses.

---

> ### Author Response · Authors · 2025-11-27
>
> > The main weakness is that there are no real-world experiments. This means the paper cannot show if the simulation is actually useful in real applications, which makes it less convincing for manipulation research.
> - As noted in our author response to all reviewers, where we report how policies trained on our generated data perform when deployed in real robot.
>
> > Another weakness is that the idea of using VLMs for simulation setup is no longer new (unlike GenSim or RoboGen). Many researchers now question the value of VLM-generated simulations. Reviewers may expect a solid and practical simulation benchmark such as THOR or RoboCasa. The authors are encouraged to build a far more diverse and convincing VLM-based simulation benchmark that includes extensive real-world validation. For the tasks in this paper, setting them up manually would be easier and more controllable than using prompts. From the policy learning point of view, compared to recent studies, it is hard to say that subtask decomposition, motion planning, or freezing degrees of freedom are contributions. If the authors want to highlight policy learning, they should provide more insights, compare with state-of-the-art methods and include real-world demonstrations.
> - We appreciate the reviewer’s perspective on recent trends in VLM-based simulation generation. However, we would like to clarify the goal of our work. Building a general-purpose simulation benchmark is not the primary intention of this paper. Unlike prior efforts such as THOR or RoboCasa, which aim to provide standardized evaluation environments, our focus is fundamentally different:
> we aim to generate large-scale training data for dexterous-hand manipulation without human intervention.
> - Our core contribution lies in leveraging VLM/LLM priors—acquired during pretraining—to automatically propose scenes, generate dexterous-hand trajectories, and construct diverse manipulation data at scale. This addresses a key challenge in dexterous manipulation: compared to grippers, dexterous hands require significantly more diverse, fine-grained, and high-quality demonstrations, whose manual collection is extremely costly. Therefore, automation of data generation—not benchmark design—is the central problem we intend to solve.
> - While many prior works question the utility of VLM-generated simulations, our sim-to-real experiments (added in the revised version) demonstrate that the data from our pipeline is indeed effective for training policies that transfer to real hardware. This supports the practical value of the proposed generative simulation pipeline beyond purely synthetic evaluation.
>
>
> > The results in Table 1 are also not convincing, and more experiments are recommended.
>
> 1. In total, our pipeline generated 50 distinct tasks, each with different objects, layouts, and manipulation requirements.
> 2. Object Asset Diversity.
>     Our pipeline incorporates a large and diverse set of manipulable assets from multiple well-established datasets, including:
>     - PartNet-Mobility: 1,284 articulated objects
>     - YCB: 78 everyday objects
>     - RoboTwin: 73 task-oriented objects
> 3. The 50 generated tasks encompass more than six distinct skill families, including: grasping, placing, opening articulated objects, in-hand rotation, closing articulated objects, and interact with articulated objects.
> 4. Distribution of number of stages
>     | **Stage Count** | 1 | 2 | 3 | 4  | 5  | 6 | 7 | 8 |
>     | --------------- | - | - | - | -- | -- | - | - | - |
>     | **# of Tasks**  | 2 | 3 | 9 | 16 | 10 | 3 | 2 | 5 |

---

### Author Response · Authors · 2025-11-27
**Sim-to-Real Results**

We would like to reiterate that the primary goal of our pipeline is **data generation**, and sim-to-real is *not* the central focus of our contribution. However, given that multiple reviewers raised questions regarding the practical usefulness of LLM/MLLM-based generative simulation for dexterous manipulation, we conducted additional sim-to-real experiments over the past two weeks to further validate the applicability of our approach. We appreciate the reviewers’ concerns, and we hope that these results help establish confidence in the proposed generative pipeline.

Below, we provide the details and outcomes of these experiments:

We adopt the ACT policy model [1] and combine simulation-generated trajectories from our pipeline with a small number of real-world demonstrations, following standard sim-to-real practices.

---

**Task 1: Close the Laptop**

* **Training data:**

  * 100 simulation trajectories generated using our pipeline
  * 5 real-world demonstrations

* **Real-world performance:**

  * **4/5 success rate (80%)**
* **Videos:**

  * Available at: [https://sites.google.com/view/gendexhand](https://sites.google.com/view/gendexhand)

This task requires precise alignment between the dexterous hand and the laptop lid, validating the usefulness of our generated data for physical deployment.

---

These results show that our generative simulation pipeline can successfully support sim-to-real transfer, even with a limited amount of real-world data. Although sim-to-real is not the main contribution of the paper, we acknowledge the reviewers’ concerns and are actively conducting additional real-robot experiments. We will provide further sim-to-real results during the remainder of the rebuttal period to strengthen the empirical support for our pipeline.

[1] Learning Fine-Grained Bimanual Manipulation with Low-Cost Hardware

---

### Meta-Review · Area_Chair_nKLi · 2026-01-07

**Summary:**

This paper studies generating data for dexterous hands in simulation. This is achieved via a pipeline composed of LLMs, VLMs, task decomposition, and RL.

All reviewers raised concerns about the lack of real-world validation. In response, the authors argued that sim2real is not a focus of the work. However, the AC agrees with the reviewers that without real-world validation, the usefulness of this work is unclear. The authors provided a new "close laptop" experiment during the rebuttal. The AC did not find this experiment compelling enough to validate the pipeline. Specifically, the details of this experiment are very light and the task itself is very simplistic.

Apart from this concern, the authors did a good job clarifying the issues regarding task diversity raised by multiple reviewers. However, the AC feels that the lack of real-world validation remains a major concern, and the paper cannot be accepted in its current form. Hopefully, the authors can address this in a future version.

**Reviewer Concerns:**

Concerns outstanding:
- Real world validation of the method

Concern addressed:
- Task diversity

**Reviewer Scores:**

Reviewer NbQh:  Keep the same or raise from 2 to 4
Reviewer MgvZ: Keep the same or raise from 2 to 4
Reviewer DQ93: Keep the same or raise from 2 to 4
Reviewer B3uT: Keep the same

---

### Decision · Program_Chairs · 2026-01-26

Reject